# Adoption of Robotic Core Technology in Minimally Invasive Lung Segmentectomy: Review

**DOI:** 10.3390/jpm12091417

**Published:** 2022-08-30

**Authors:** Takashi Eguchi, Kentaro Miura, Kazutoshi Hamanaka, Kimihiro Shimizu

**Affiliations:** Division of General Thoracic Surgery, Department of Surgery, Shinshu University School of Medicine, 3-1-1 Asahi, Matsumoto 390-8621, Japan

**Keywords:** lung segmentectomy, minimally invasive surgery, robotic-assisted thoracic surgery, surgical skills

## Abstract

A recent randomized trial demonstrated the survival superiority of lung segmentectomy over lobectomy in patients with early stage, small-sized lung cancer. Hence, there is a pressing need for thoracic surgeons to gain familiarity with lung segmentectomy. However, lung segmentectomy, especially via minimally invasive surgery, is a technically challenging thoracic surgical procedure. The robotic surgery platform helps surgeons to improve their operative performance based on its core technological features: improved dexterity, precision, and visualization. Herein, we have discussed the key issues related to robotic lung segmentectomy, explicitly focusing on the technical features of complex segmentectomy under difficult conditions. We have also introduced our preferred surgical strategy for robotic lung segmentectomy with specific maneuvers.

## 1. Introduction

In this era of precision medicine, surgeons should provide “personalized” surgery tailored to individual patients based on the oncological features of lung tumors, hilar anatomy, functional background, and surgical tolerance [1].

Thoracic minimally invasive surgery (MIS), including video-assisted thoracic surgery (VATS) and robotic-assisted thoracic surgery (RATS), has increasingly been performed, with growing evidence of better postoperative pain control [2], shorter hospitalizations, and lower risk of postoperative complications [3] and non-cancer-specific mortality [4]. During MIS, unexpected conversion to open thoracotomy is occasionally unavoidable and is associated with an increased risk of postoperative complications and mortality [5,6,7].

Recently, a large, multi-institutional, prospective randomized trial assessing the outcomes of segmentectomy and lobectomy for small lung tumors (≤2 cm) (JCOG0802/WJOG4607L) demonstrated that the overall survival of patients who underwent segmentectomy was significantly superior to that of patients who underwent lobectomy [8]. Despite a higher local recurrence in the segmentectomy group compared with the lobectomy group, there was no difference in the hazard of lung cancer death between the two groups. More importantly, death from other cancers, respiratory disease, and cerebrovascular disease occurred less frequently in the segmentectomy group than in the lobectomy group. 

The robotic surgery platform helps surgeons to improve their operative performance based on its core technological features: improved dexterity, precision, and visualization. RATS lung segmentectomy is one of the procedures that meet the requirements for minimal invasiveness and lung volume preservation [9], which potentially supports an increasing use of the procedure [10,11]. However, the procedure has been considered to be technically challenging and has not been standardized. Although several studies have discussed the techniques of robotic lung segmentectomy, they mainly focused on sequences of the procedure and/or its port placement [12,13,14,15,16]. However, based on our surgical experience in RATS segmentectomy, it is also important for surgeons to learn and gain expertise in maximizing the robotic core technology for segmentectomy specific maneuvers with an optimal use of robotic devices and a precise anatomical understanding with accurate visualization.

Herein, to address the pressing need for thoracic surgeons to manage patients in a personalized way based on individual tumor size, location, and stages using both an oncologically and a functionally optimal treatment strategy, we have summarized the published evidence and discussed the key issues related to RATS lung segmentectomy. We have specifically focused on the technical features of the procedure based on robotic core technology and introduced our preferred surgical strategy with specific maneuvers. 

## 2. Technical Features of Robotic Lung Segmentectomy

MIS lung segmentectomy has been found to be associated with better outcomes compared with segmentectomy via open thoracotomy [17]. Several single-center reports and population-based studies have demonstrated that the short- and long-term outcomes following RATS segmentectomy are clinically feasible and equivalent to those of the VATS segmentectomy [11,13,16,18]. Although there has been no apparent superiority of RATS in terms of reported post-segmentectomy outcomes compared to VATS, the number of RATS procedures performed in segmentectomy has increased significantly [11]. In contrast, the proportions of the use of VATS and open thoracotomy have decreased [11]. Potential reasons for the increase in RATS segmentectomy include its advantageous technical features.

### 2.1. Unique Techniques in Lung Segmentectomy and the Advantage of the Use of RATS

Lung segmentectomy is associated with technical challenges because it requires a deep hilar dissection to identify the segmental branches that need to be divided (or preserved) and a division of multiple intersegmental planes [19]. Therefore, segmentectomy requires unique dissecting techniques related to segmentectomy specific procedures. Table 1 demonstrates segmentectomy specific procedures with frequently required maneuvers during lung segmentectomy. 

Although the procedural sequences of lobectomy and segmentectomy are similar, segmentectomy requires additional procedures, including the following: (1) identification of and dissection along the intersegmental veins to appropriately divide the central intersegmental plane; (2) identification and division (or preservation) of the segmental and/or subsegmental branches of the vessels and bronchi; and (3) identification and division of the peripheral intersegmental plane [9,19].

In general, a surgical robot platform provides surgeons with improved dexterity, precision, and binocular visualization, which are considered robotic core technologies. We developed a three-step strategy for RATS segmentectomy with a proposal of step-specific maneuvers utilizing robotic core technologies (Figure 1) [19]. Through our experiences with RATS segmentectomies using the three-step strategy, it is apparent that RATS is well suited to perform the unique maneuvers required for segmentectomy.

### 2.2. Choice of Dissecting Instruments for Lung Segmentectomy: Spatula Versus Bipolar Forceps

Most thoracic surgeons opt to use a dissecting cautery device (e.g., Cautery Spatula^®^, Cautery Hook^®^, Maryland Bipolar Forceps^®^, Long Bipolar Grasper^®^, Curved Bipolar Dissector^®^, etc.) in the right-hand arm and a grasping device with/or without a cauterizing function (e.g., Fenestrated Bipolar Forceps^®^, Cardier Forceps^®^, etc.) in the left-hand arm. The choice of robotic instruments from multiple options, especially in the right-hand dissecting devices, depends on surgeons’ preferences and intraoperative conditions. However, fundamental differences in dissecting devices between monopolar cautery and bipolar forceps can significantly affect surgical performance during segmentectomy. Table 1 demonstrates the performance of right-hand dissecting instruments during each segmentectomy specific procedure separately for monopolar cautery and bipolar forceps. 

To complete these segmentectomy specific procedures with appropriate hemostasis, we prefer to use Maryland Bipolar Forceps^®^ in the right-hand arm during segmentectomy in our institution. We proposed a three-step strategy for robotic lung segmentectomy by illustrating specific robotic maneuvers using Maryland Bipolar Forceps^®^ in the right-hand arm and Fenestrated Bipolar Forceps^®^ in the left-hand arm [19].

### 2.3. Robotic Versus Hand-Held Staplers during Robotic Lung Segmentectomy 

During robotic lung segmentectomy, surgical staplers are utilized to divide the segmental vascular and bronchial branches and the intersegmental planes. In RATS lung resection, surgeons can elect to use either robotic staplers or hand-held staplers (in da Vinci *S* or *Si* systems, only hand-held staplers are available). A previous study reported a lower risk of postoperative complications after the use of a robotic stapler compared with a hand-held stapler for lung lobectomy [20]. In general, robotic staplers would surpass hand-held staplers in terms of the maneuverability due to the completely wristed, omnidirectional articulation. In addition, many surgeons prefer the autonomous, direct maneuverability of the robotic stapler from the console. Potential drawbacks of the robotic stapler would be the requirement of a 12-mm robotic cannula instead of an 8-mm cannula and the unavailability of narrow vascular staplers, which can be a disadvantage of robotic staplers during segmentectomy in patients with multiple small pulmonary artery segmental or sub-segmental branches that are suitable for narrow vascular staplers. Table 2 demonstrates functional aspects of robotic and hand-held staplers during RATS. 

## 3. Topographic Anatomy Based Surgical Sequences for Personalized Segmentectomy

In contrast to lung lobectomy, which consists of only five patterns, there is a wide range of patterns of lung segmentectomies, including sub-segmentectomies and combined segmentectomies. In individual patients, appropriate segmentectomy with adequate surgical margins and an acceptable loss of pulmonary function should be planned preoperatively based on the size and location of the tumor and the patient’s background characteristics. 

In general, the RATS approach has an advantage in obtaining better surgical visualization due to the magnified visualization using its binocular scope system and the steady retraction using its assistant arm (third arm). However, RATS would not be fit for frequent, dynamic changes of surgical fields, which is a drawback of RATS compared with VATS, and the open approach and may cause the false recognition of complex segmental anatomy during surgery.

During lung segmentectomy, thoracic surgeons should be confident in the following for appropriate intraoperative decision-making: (1) differentiation between segmental branches being divided and preserved; (2) appropriate sequential order for dividing the segmental branches; and (3) adequate margin distance. These may be challenging for thoracic surgeons since there is a wide variety of segmental/sub-segmental branching patterns of the pulmonary vessels and bronchi with frequent anomalies. Therefore, precise preoperative planning and intraoperative navigation are crucially based on individual segmental topographic anatomy.

### 3.1. Procedural Sequences Based on Topographic Segmental Anatomy

Although there are infinite variations in hilar segmental anatomical patterns, several anatomical classifications of the lung hilum have been reported [21,22,23]. Our colleague found that three-dimensional computed tomography (3D-CT) imaging was useful for measuring anatomical branching patterns of pulmonary vessels and bronchi more easily and precisely than the traditional way to research anatomy using cadaver organs [22,23,24,25,26]. More importantly, anatomical analysis using 3D-CT imaging has been utilized to create surgery oriented anatomical classifications by which thoracic surgeons can preoperatively develop a plan of procedural sequences. Our group previously developed a simplified 3D anatomical model of the right upper lobe, focusing on appropriate intraoperative access to the intersegmental veins, which are an essential topographic landmark for lung segmentectomy [23]. In this study, three standardized surgical approaches to the intersegmental veins (anterior, interlobar, and posterobronchial approaches) during right upper lobe segmentectomy were proposed. The choice of one approach among the three depends on the patient’s anatomical classification pattern, which was developed using a large cohort of consecutive patients with preoperative contrast CT scans. The clinical outcomes of consecutive patients who underwent right upper lobe segmentectomy using the topographic 3D anatomy based surgical approaches were reported [26].

### 3.2. Preoperative Simulation Based on Segmentectomy Procedure Sequence-Specific 3D Images

The 3D-CT imaging has been helpful for preoperative simulation and intraoperative navigation during segmentectomy [25,27,28]. However, previous 3D-CT reconstruction software has had potential drawbacks such as time-consuming reconstruction processes, the requirement of a contrast-enhanced CT scan, and a lack of realistic simulation images [29]. Recently, next-generation, surgeon-oriented 3D-CT reconstruction platforms have become available [9,30,31], through which surgeons can develop not only traditional 3D-CT reconstruction automatically, but also modify the developed images semi-automatically and create procedure-specific 3D-CT imaging analysis. 

In our institution, we utilized a next-generation 3D-CT reconstruction platform (Revoras^®^, Ziosoft, Inc., Tokyo, Japan) that has a segmentectomy planning function. This planning provides surgeon-oriented specific 3D views showing the hilar structures being divided with stumps and the dissected intersegmental planes. Before the preoperative conference, the primary surgeon or trainees prepare sequential images with vascular or bronchial stumps in a surgery simulated view, which will then be presented at the preoperative conference with a demonstration of topographic segmental anatomical features, including branching patterns, presence of anomalies, and order of surgical sequences using sequential images (Figure 2).

## 4. Personalized Strategies and Techniques for Challenging Conditions

As JCOG0802/WJOG4607L demonstrated the survival superiority of lung segmentectomy over lobectomy, thoracic surgeons should have the incentive to perform segmentectomy in various patients even under challenging conditions. These challenging conditions may include: segmentectomy at the lung base; rare segmentectomy such as medial-basilar segmentectomy; pulmonary artery adherent lymph nodes; and deep, small tumors requiring tumor localization and resection. We describe the technical feasibility of RATS in the above-mentioned challenging conditions except for the latter, for which we previously reviewed literature and proposed our strategy using radiofrequency identification technology [9].

### 4.1. Complex Segmentectomy at the Lung Base: Lung Base-Flip Approach

In general, single segmentectomies at the lung base are more technically challenging than other lung segmentectomies because of their anatomical complexity and difficulty in identifying the intersegmental planes [9,22]. To overcome these technical challenges, we proposed the lung base-flip approach for robotic basilar segmentectomies (the 58th Annual Meeting of the Society of Thoracic Surgeons, 29–30 January 2022; the 102nd Annual Meeting of the American Association of Thoracic Surgery, 14–17 May 2022). Appendix A shows a 3D-CT demonstration of the lung base-flip approach. 

The advantages of this approach include better access to the basilar intersegmental veins at the beginning of hilar dissection, negligible impact of the fissure completeness, and good compatibility with the “look-up” view of conventional RATS. 

### 4.2. Right Medial-Basilar (S^7^) Segmentectomy and Basilar Segmentectomies Preserving the Medial-Basilar Segment

The medial-basilar segment (S^7^) is located in the medial portion of the right lower lobe, which exists only in the right lung and is the smallest segment among all lung segments [24,32]. Our group demonstrated the technical feasibility of right medial-basilar segmentectomy via VATS [32]. A robotic approach is also feasible for the right medial-basilar segmentectomy (Appendix A). 

Although the incidence of encountering patients with a small tumor in the right medial-basilar segment is not high, thoracic surgeons who perform segmentectomies in the right-lower lobe should be familiar with the segmental anatomy of the medial-basilar segment, which is also essential for anatomically correct segmentectomies in the right lung base, including anterior-basilar (S^8^), lateral-basilar (S^9^), and posterior-basilar (S^10^) segmentectomies, as well as their combinations. It is important to sufficiently divide the central intersegmental plane between the medial basilar segment and other segments to obtain a wide opening for subsequent deep hilar dissection at the beginning of the procedure.

### 4.3. Segmentectomies in Patients with Pulmonary Artery Adherent Lymph Nodes

The presence of a pulmonary artery adherent lymph node, which is a lymph node firmly adherent to the wall of the pulmonary artery without a loose dissection plane due to hilar inflammation and/or anthracofibrosis, causes difficulty in hilar dissection and increases the risk of intraoperative catastrophes such as pulmonary arterial injury and/or conversion to thoracotomy during MIS [7,33]. In our previous study, we developed a risk stratification model for the presence of pulmonary artery adherent lymph nodes based on preoperative bronchoscopy and CT scan findings [7]. Based on the risk stratification, we developed a patient selection protocol for MIS lung resection [7]. In high-risk patients with pulmonary artery adherent lymph nodes, we discuss the operative risk with patients to determine whether we should perform open thoracotomy instead of MIS or proceed with MIS by adopting measures such as assigning a senior surgeon as the primary surgeon, preparing for blood transfusion, and administering epidural analgesia [7]. It is crucial for surgeons not to hesitate elective conversion to thoracotomy if hilar dissection is difficult to perform in a minimally invasive method. 

Robotic core technologies (improved dexterity, precision, and visualization) may help surgeons to provide patients with pulmonary artery adherent lymph nodes with safe, secure, and minimally invasive lung resection. Although the da Vinci system does not provide tactile sensation, we suggest that surgeons can appropriately evaluate the presence of pulmonary artery adherent lymph nodes intraoperatively through its improved visualization. In addition, skills for robotic segmentectomy might be useful for dissecting and dividing the pulmonary artery distal to the adherent portion. Appendix A demonstrates our surgical strategy and techniques in a patient with pulmonary artery adherent lymph nodes.

## 5. Limitations

Several descriptions in this review were based on the authors’ surgical experiences and were not supported by scientific evidence. The first author’s robotic experiences were based on a current practice as an attending surgeon in Shinshu University Hospital, Matsumoto, Japan, using the da Vinci *Si* system and a previous thoracic surgical fellowship training in the Thoracic Surgical Oncology Fellowship program in the Memorial Sloan Kettering Cancer Center (MSK), New York, USA, for the 2018/2019 academic year using the da Vinci *Xi* system. 

## 6. Conclusions

Here, we have discussed robotic lung segmentectomy with a particular focus on technical features. Based on recent evidence demonstrating segmentectomy’s survival superiority over lobectomy, thoracic surgeons should acquire expertise in complex segmentectomy skills, which are required for personalized surgical treatment in patients with early-stage lung cancer.

## 7. Future Directions

A higher prevalence of lung segmentectomy performed with adequate skills will contribute to better prognoses in patients with early-stage lung cancer. Robotic surgery platforms will help thoracic surgeons overcome technical challenges and may further improve quality and surgical outcomes beyond the current realm of possibility.

## Figures and Tables

**Figure 1 jpm-12-01417-f001:**
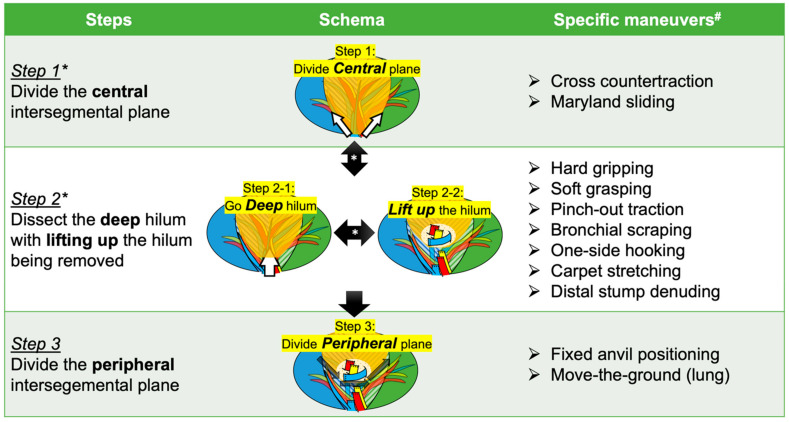
Three-step strategy for robotic lung segmentectomy. * Steps 1 and 2 can be repeated before proceeding to Step 3. ^#^ Specific maneuvers are illustrated in our previous publication [19].

**Figure 2 jpm-12-01417-f002:**
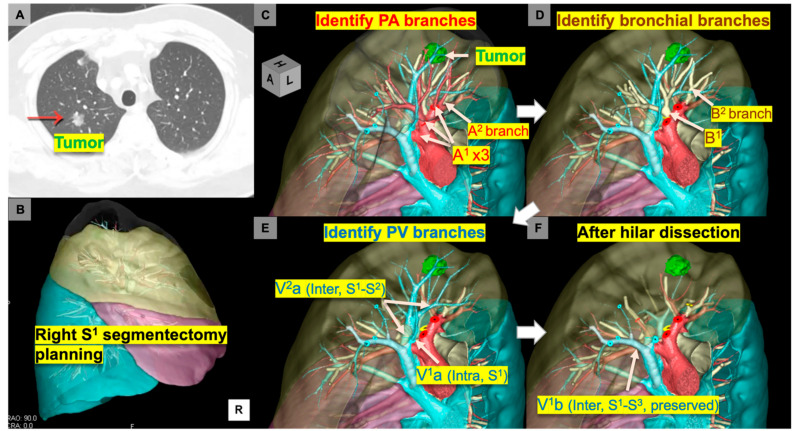
Sequential 3D images simulating right apical (S^1^) segmentectomy. (**A**) A computed tomography image shows a 1.8-cm subsolid nodule in the right S^1^. (**B**) A lateral view of 3D simulating image. (**C**–**F**) Surgery-simulated 3D views for identifying PA branches (**C**), bronchial branches after dividing PA branches (**D**), PV branches after dividing bronchial branches (**E**), and hilar view after dividing segmental branches to be removed (**F**). Inter, intersegmental vein; Intra, intrasegmental vein; PA, pulmonary artery; PV, pulmonary vein; S, segment.

**Table 1 jpm-12-01417-t001:** Segmentectomy specific procedures with frequently required maneuvers in RATS lung segmentectomy.

Segmentectomy Specific Procedures	Frequently Required Maneuvers	Suitability of Dissecting Devices	
		Monopolar Cautery ^a^	Bipolar Forceps ^b^
** *Deep hilar dissection* **			
Hemostasis in the deep hilum	Primary hemostasis by sponge compression	(-)	(++)
	Hemostasis by pin-point cauterization	(+)	(++)
	Hemostasis by half-open cauterization using Fenestrated Bipolar	N/A	N/A
Obtaining operative field with traction in the deep hilum	Hard gripping of tissues surrounding a lymph node using grasping forceps ^#^	N/A	N/A
	Soft grasping of a whole lymph node using grasping forceps ^#^	N/A	N/A
	Pinch-out traction maneuver ^c#^	N/A	N/A
Identification and division of segmental branches	Circumferential dissection of segmental branches	(+)	(++)
	Scraping tissues off the bronchus ^#^	(+)	(++)
	Cauterization of small vascular branches	(+)	(++)
	Ligation of branches	(-)	(++)
	Clipping of branches	N/A	N/A
	Distal stump denuding maneuver (for identifying and dissecting the corresponding branch ^d^) ^#^	(+)	(+)
Lifting up the hilum being removed	Distal stump denuding maneuver (for separating the hilum being removed from the hilum being preserved) ^#^	(+)	(+)
** *Division of the intersegmental plane* **			
Identification and division of the central intersegmental plane	Dividing the “visible plane with sparse tissue” along the intersegmental vein	(++)	(+)
	Dividing the “invisible plane with dense tissue” along the intersegmental vein	(+)	(++)
	Cross-counter traction maneuver ^#^	N/A	N/A
Identification of the peripheral intersegmental plane	Inflation–deflation	N/A	N/A
	Perfusion–non-perfusion	N/A	N/A
Division of the peripheral intersegmental plane	Stapling with fixed anvil positioning and move-the-lung maneuvers ^#^	N/A	N/A
	Dividing with cautery along the inflation-deflation border	(++)	(+)

^a^ Monopolar cautery includes Cautery Spatula^®^ and Cautery Hook^®^. ^b^ Bipolar forceps include Maryland Bipolar Forceps^®^, Long Bipolar Grasper^®^, or Curved Bipolar Dissector^®^. ^c^ Spreading tissues using grasping devices such as Fenestrated Bipolar Forceps^®^ or Cardier Forceps^®^. ^d^ A corresponding branch represents a segmental artery or bronchus that accompanies its corresponding segmental bronchus or artery (e.g., A^1^ is a corresponding artery of B^1^). Definitions of marks: (-), unsuitable; (+), usable but there would be other suitable instruments; (++), suitable. N/A, not applicable; RATS, robotic-assisted thoracic surgery. ^#^ These maneuvers are illustrated in a published tutorial video [19].

**Table 2 jpm-12-01417-t002:** Functional aspects of robotic and hand-held staplers during robotic-assisted thoracic surgery.

Functional Aspects	Robotic Stapler	Hand-Held Stapler
Autonomous control of stapler by console surgeon	Full control (direct control)	None (indirect control via bedside assistant)
Articulation (direction)	Completely wristed articulation (omnidirectional articulation)	Only horizontal articulation
Articulation (degree)	Up to 120° cone of articulation *	Up to 90°–110° cone of articulation
Narrow (small diameter) vascular stapler	Not available	Available
Requirement of the use of 12-mm robotic canula	Yes	No
Availability for each da Vinci system	Only available in *X* or *Xi* systems	All da Vinci systems (*S*, *Si*, *X*, *Xi*) are available

* Available in SureForm^®^ stapler.

## Data Availability

Not applicable.

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
