# Peer review of "Adoption of Robotic Core Technology in Minimally Invasive Lung Segmentectomy: Review"

_jpm, 2022, doi:10.3390/jpm12091417_

Round 1
Reviewer 1 Report
The authors have discussed key issues related to robotic lung segmentectomy, explicitly focusing on the technical features of complex segmentectomy. Recently, there have been a lot of reports of robotic complex segmentectomy. It is interesting to consider how to safely and reliably perform complex segmentectomy. The authors described surgical strategy for robotic segmentectomy in a specific and straightforward manner. It is difficult to describe such a strategy in the manuscript, but the strategy was well organized. However, the differences from previous reports or the robot stapler were not described well, which may to be added.
Author Response
Response to Reviewer #1
We thank reviewer #1 for the comments. Please find our specific responses below.
- The authors have discussed key issues related to robotic lung segmentectomy, explicitly focusing on the technical features of complex segmentectomy. Recently, there have been a lot of reports of robotic complex segmentectomy. It is interesting to consider how to safely and reliably perform complex segmentectomy. The authors described surgical strategy for robotic segmentectomy in a specific and straightforward manner. It is difficult to describe such a strategy in the manuscript, but the strategy was well organized. However, the differences from previous reports or the robot stapler were not described well, which may to be added.
Response: We thank the reviewer for the comment. We agree with reviewer #1 regarding the lack of description on the differences from previous reports and the use of robotic stapler. Several previous papers discussing techniques of robotic lung segmentectomy described sequences of the procedure and/or its port placement. However, during robotic lung segmentectomy, it is also important for surgeons to learn and expertise in maximizing robotic core technology application using robotic-specific devices appropriately with precise anatomical visualization according to segmentectomy-specific sequences, which was not covered by previous publications; we have attempted to describe it in this paper. We have added and modified the following sentences in the Introduction section.
‘The robotic surgery platform helps surgeons to improve their operative performance based on its core technological features: improved dexterity, precision, and visualization. RATS lung segmentectomy is one of the procedures that meet the requirements for minimal invasiveness and lung volume preservation [9], which potentially supports an increasing use of the procedure [10, 11]. However, the procedure has been considered to be technically challenging and has not been standardized. Although several studies have discussed the techniques of robotic lung segmentectomy, they mainly focused on sequences of the procedure and/or its port placement [12-16]. However, based on our surgical experience in RATS segmentectomy, it is also important for surgeons to learn and expertise in maximizing the robotic core technology for segmentectomy-specific maneuvers with optimal use of robotic devices and precise anatomical understanding with accurate visualization.’
(Introduction, pages 1–2, lines 39–52)
Regarding the stapler, robotic staplers would surpass hand-held staplers in terms of the maneuverability by a console surgeon instead of a bedside assistant and the completely wristed articulation of the stapler. Potential disadvantages of robotic staplers compared with hand-held staplers would include a requirement of a 12-mm robotic port instead of an 8-mm port. We have added a paragraph and a table on the comparison between robotic staplers and hand-held staplers in our second section (technical features of robotic lung segmentectomy) as follows.
‘2.3. Robotic versus hand-held staplers during robotic lung segmentectomy
During robotic lung segmentectomy, surgical staplers are utilized to divide the segmental vascular and bronchial branches and the intersegmental planes. In RATS lung resection, surgeons can elect to use either robotic staplers or hand-held staplers (in da Vinci S or Si systems, only hand-held staplers are available). A previous study suggested lower risk of postoperative complications after the use of robotic stapler compared with hand-held stapler for lung lobectomy [20]. In general, robotic staplers would surpass hand-held staplers in terms of the maneuverability due to the completely wristed, omnidirectional articulation. In addition, many surgeons prefer the autonomous, direct maneuverability of robotic stapler from the console. Potential drawbacks of robotic stapler would be a requirement of a 12-mm robotic cannula instead of an 8-mm cannula, and unavailability of narrow vascular staplers, which can be a disadvantage of robotic staplers during segmentectomy in patients with multiple small pulmonary artery segmental or sub-segmental branches that are suitable for narrow vascular staplers. Table 2 demonstrates functional aspects of robotic and hand-held staplers during RATS. ’
(Technical features of robotic lung segmentectomy, pages 5–6, lines 112–127)

Reviewer 2 Report
This is a very nice description of technical details about robotic segmentectomy
It is not clear the aim of the paper. In the title the Authors state that it is a review paper, but in fact it is more a decription of technical issues about robotic segmentectomy, based on a large surgical experience.
The methods have not been decribed
There are many statements and opinions expressed by the Authors that are not supported by any scientific evidence. Maybe they are correct, but in an original paper they should be better supported. As they are expressed by the Authors, they are more suitable for a bok chapter than for a scientific paper
Author Response
Response to Reviewer #2
We thank reviewer #2 for the suggestions and comments. Please find our specific responses below.
- This is a very nice description of technical details about robotic segmentectomy. It is not clear the aim of the paper. In the title the Authors state that it is a review paper, but in fact it is more a decription of technical issues about robotic segmentectomy, based on a large surgical experience. The methods have not been described
Response: We thank the reviewer for the comments. We agree with the reviewer that the aim of this paper was not clear and there was no predetermined, specified strategy for literature search. Our surgical experience in RATS segmentectomy have led us to believe that it is important for surgeons to learn and expertise in maximizing robotic core technology application utilizing robotic-specific devices appropriately with precise anatomical visualization according to segmentectomy-specific sequences. In our literature review in the Pub-Med: ((((technique) OR (skills)) AND (robotic)) AND (segmentectomy)) AND (lung), we found 114 articles, among which several papers discussed techniques of robotic lung segmentectomy mainly focusing sequences of the procedure and/or its port placement. However, during robotic lung segmentectomy, it is also important for surgeons to learn how to maximize robotic core technology using robotic-specific devices appropriately with precise anatomical visualization according to segmentectomy-specific sequences, which was not covered in previous publications, and we have attempted to describe it in this paper. We have added and modified the following sentences in the Introduction section. We did not create a methods section according to the journal’s instruction.
‘The robotic surgery platform helps surgeons to improve their operative performance based on its core technological features: improved dexterity, precision, and visualization. RATS lung segmentectomy is one of the procedures that meet the requirements for minimal invasiveness and lung volume preservation [9], which potentially supports an increasing use of the procedure [10, 11]. However, the procedure has been considered to be technically challenging and has not been standardized. Although several studies have discussed the techniques of robotic lung segmentectomy, they mainly focused on sequences of the procedure and/or its port placement [12-16]. However, based on our surgical experience in RATS segmentectomy, it is also important for surgeons to learn and expertise in maximizing the robotic core technology for segmentectomy-specific maneuvers with optimal use of robotic devices and precise anatomical understanding with accurate visualization.’
(Introduction, pages 1–2, lines 39–52)
- There are many statements and opinions expressed by the Authors that are not supported by any scientific evidence. Maybe they are correct, but in an original paper they should be better supported. As they are expressed by the Authors, they are more suitable for a book chapter than for a scientific paper.
Response: We thank the reviewer for the comment and suggestion, which we agree with. We have added a limitation section to describe the issues that the reviewer suggested.
‘Several descriptions in this review were based on the authors’ surgical experiences and were not supported by scientific evidence. The first author’s robotic experiences were based on a current practice as attending surgeon in Shinshu University Hospital, Matsumoto, Japan using da Vinci Si system and a previous thoracic surgical fellowship training in the Thoracic Surgical Oncology Fellowship program in Memorial Sloan Kettering Cancer Center (MSK), New York, USA for 2018/2019 academic year using da Vinci Xi system.’
(Page 10, lines 281–287)
